# Menstrual Cycle Patterns and the Prevalence of Premenstrual Syndrome and Polycystic Ovary Syndrome in Korean Young Adult Women

**DOI:** 10.3390/healthcare9010056

**Published:** 2021-01-07

**Authors:** Young-Joo Park, Hyunjeong Shin, Songi Jeon, Inhae Cho, Yae-Ji Kim

**Affiliations:** 1College of Nursing, Korea University, Seoul 02841, Korea; yjpark@korea.ac.kr (Y.-J.P.); hyunjshin@korea.ac.kr (H.S.); inhae05@naver.com (I.C.); 2Health Insurance Review & Assessment Service, Kangwon 26465, Korea; yeji0967@naver.com

**Keywords:** women, young adult, menstrual cycle, premenstrual syndrome, polycystic ovary syndrome

## Abstract

Menstruation is one of the important indicators of reproductive health. Therefore, in order to improve the reproductive health of women in puberty and early adulthood, it is necessary to investigate menstrual health and symptoms. This cross-sectional descriptive correlational study was conducted to identify young women’s menstrual cycle patterns, prevalence of Premenstrual Syndrome (PMS), Polycystic Ovary Syndrome (PCOS) and the relationships of health-related factors according to menstrual regularity and PCOS. 462 women participated in the first phase of the study and completed the menstrual health and health-related behaviors questionnaire. In the second phase, 88 women with irregular menstruation in phase one had blood tests taken and body composition measured. As a result, Menarche was slightly later in irregular menstruation group. Women with regular menstruation had a mean number of 11.7 menstrual cycles over the past year, 93.0% of them reported a normal menstruation cycle frequency (21–35 days), 95.2% reported a normal duration (2–7 days) and 55.9% of participants had heavy menstrual bleeding. In the irregular menstrual group, there were higher percentages of underweight and obese women as well as more women experiences weight and diet changes. The estimated prevalence rates of PMS and PCOS were 25.5%, 5.2% respectively. This study provides updated basic data about menstrual health among Korean young women but more extensive and sophisticated studies are needed in the future.

## 1. Introduction

Reproductive health addresses the reproductive processes, functions and system at all stages of life. Therefore, reproductive health implies the ability to lead a responsible and satisfactory sexual life and the freedom to decide when to have children and how many children [1]. Menstruation is one of the important indicators of reproductive health and menstrual health is evaluated not only by the parameters of the menstrual cycle patterns but also by menstrual symptoms including premenstrual disorder (PMD). The menstrual cycle appears in a series of feedback processes of hypothalamus–pituitary–ovarian (HPO) axis. The typical menstrual cycle pattern is 28 ± 7 days with menstrual flow lasting 4 ± 2 days and blood loss averaging 20 to 60 mL. Menstrual cycle intervals vary among women and often for an individual woman at different times during her reproductive life [2].

Changes in menstrual cycle patterns and volume of menstrual fluid are defined as abnormal uterine bleeding (AUB) and AUB is classified into two categories: irregular and heavy menstrual bleeding [3]. The reported prevalence of AUB varies according to populations and to the definition of AUB being used but it is widely known that at least 3 to 30% of women have experienced it [4].

The major causes of AUB differ according to the stages of life; in puberty/early adulthood, ovulation disorder may be the cause of AUB and the bleeding patterns vary widely from amenorrhea to irregular, heavy bleeding. During this period, ovulation disorder can be ascribed to physiological causes such as immature HPO axis for several years after menarche but it also can be related to mental stress and eating disorders. AUB can also be associated with polycystic ovary syndrome (PCOS), endocrinopathies such as thyroid disease, coagulation disorders and cancer [5]. In other words, menstrual abnormalities outside of the normal range in women in puberty/early adulthood may be related to temporary causes such as psychological or physical stress, while chronic anomalies may be associated with pathologic causes such as PCOS, endometriosis, hypogonadism or cancer [6].

In particular, PCOS is an endocrinopathy that should be considered a priority diagnosis in women in this developmental period, because the prevalence rate is 10%. In addition, there are various diagnostic criteria to be aware of, including hyperandrogenism, rare (or no) ovulation and polycystic ovary. PCOS increases susceptibility to other disorders including infertility, obstetric complications, type Ⅱ diabetes, cardiovascular disease, mood disorders and eating disorders [7]. In the Nurses’ Health Study, 10.9% of women aged 20–35 years had irregular menstruation and 4.3% had severe irregular menstruation, both of which seem to be related to PCOS [8]. Thus, women in puberty/early adulthood are at high risk for concomitant pathologies such as PCOS and AUB, as well as for more health risk behaviors such as sexual activity, poor diet/eating disorders and alcohol/substance abuse [9].

In addition, women experience PMD symptoms cyclically and repeatedly and the disorder affects women’s everyday lives. PMD is divided into core and variant, with premenstrual syndrome (PMS) and premenstrual dysphoric disorder (PMDD) being classified as core PMD. PMS is a form of PMD that manifests as a variety of physical, mental or behavioral symptoms repeatedly during the luteal phase of the menstrual cycle and disappears within the first few days after the beginning of menstruation and PMDD is a severe form of PMS [10]. However, among studies on PMS/PMDD in Korea, only a very limited number used tools to identity clinically significant PMS/PMDD according to the DSM-IV or the diagnosis criteria of the American Congress of Obstetricians and Gynecologists [11,12].

Therefore, in order to improve the reproductive health of Korean women in puberty and early adulthood, it is necessary to secure updated basic data about their menstrual cycle patterns, the status of clinically significant PMS and PCOS and their related factors. In other words, it requires data about a specific demographic group of women to determine the normal range of menstrual cycle patterns of the demographic group concerned, as the menstrual cycle patterns of women in this period are complicatedly related with not only physiological factors like immaturity of HPO axis and medical conditions but also socioeconomic and behavioral factors [13]. In order to identify the symptoms and status of clinically significant PMS/PMDD, it is necessary to conduct a study using PMS tools that are validated and recommended and to secure updated basic data about the status and related factors of PCOS, which should be considered first in terms of chronic menstrual problems in women of this period.

In this regard, this study was designed first to investigate menstrual cycle; second, to identify the symptoms and prevalence of clinically significant PMS; third, to estimate the prevalence of PCOS; and lastly, to identify the relationship between health-related behaviors (smoking, drinking, eating habits and nutrients intake), body composition and blood indexes (total testosterone (T), sex hormone binding globulin (sHBG), fasting blood sugar (FBS) and insulin) according to menstrual cycle regularity and the presence or absence of PCOS.

## 2. Materials and Methods

### 2.1. Study Design

The first phase of this study was a cross-sectional descriptive correlational investigation of menstrual cycle patterns and the prevalence of clinically significant PMS, followed by estimating their relationships with health-related behaviors. In the second phase, the prevalence of PCOS was estimated based on the findings from phase one of the study in addition to PCOS’s relationship with body composition and blood indexes.

### 2.2. Setting and Samples

The participants of this study as a convenience sample consisted of young adult women aged 18 to 29 years who were undergraduate and graduate students at K University, excluding women who were using estrogen and progesterone-containing hormone drugs. To recruit the participants for the first phase of the study, the online and offline advertisement were used from 28 August to 22 December 2017. When candidates revealed their intention to participate through e-mail, QR code or URL connection, they were given online a copy of the informed consent form. A total of 522 women signed these forms and 470 women responded to the questionnaires. After excluding incomplete respondents, 462 women were selected for the final analysis.

For the second phase of the study, 132 women who had irregular menstruation defined as experiencing it fewer than 10 times per year based on previous study [14] were selected from the phase-one participants. During the phase-one recruitment, it was specified in the notices that candidates could be possibly selected for the second-phase study. During the online survey in phase one, it was informed participants that they were selected for phase two. 88 of them expressed their intention to participate voluntarily and they were the participants in the final study.

### 2.3. Ethical Considerations

The study was conducted after obtaining the approval of the K University Internal Review Board (IRB No. 1040548-KU-IRB-17-112-A-1). The participants for the first-phase study were recruited by placing recruitment notices online and offline. After those candidates who wanted to participate were informed online of the purpose of the research, the disposal of the collected data and the voluntary participation and rejection of participants, they signed the written form of consent to participate in the research. Among the phase one study participants, women who met the requirements for phase two were selected and informed them of the purpose of that study, participation method and so on. The blood tests were scheduled in agreement with those who wanted to participate and after the women signed the final consent form, the second phase of the study was conducted. It was explained to all participants that they could drop out of the research at any time if they did not want to participate any longer and that there would be no penalty for withdrawing.

### 2.4. Measurements and Instruments

#### 2.4.1. Phase One

Menstrual Cycle Patterns: menstrual cycle patterns were assessed by cycle frequency, duration, regularity and volume. The frequency was defined as the interval between menstrual cycles, duration as the period from the beginning to the end of menstrual bleeding and regularity as variations of menstrual cycle length. The questionnaire to measure the menstrual cycle patterns consisted of self-report questions about the women’s menstrual periods over the previous one year. Menstrual flow volume was measured using the pictorial blood assessment chart (PBAC) to assess menstrual blood loss [11,15]. The PBAC was designed to visualize and record the volume of bleeding in pads, tampons and blood clots and scored. The higher score, the greater the volume of menstrual flow and higher than 100 points is defined as menorrhagia.

Menstrual Symptoms: Menstrual symptoms were assessed using the Daily Record of Severity of Problems (DRSP) [11,16]. This tool takes questions directly from the DSM for the diagnosis of PMS/PMDD: 21 questions (Criteria A) about psychological (depression, anxiety, mood swings, anger, social withdrawal, concentration difficulty, lethargy, appetite changes, sleep, control) and physical (breast, headache, joint/muscle pain) symptoms and 3 additional questions (Criteria B) about functional impairment. The DRSP was originally developed to measure symptoms on a daily basis in a prospective way but because of its inconvenience, PMS was determined based on the scores that were retrospectively measured on the first day of menstruation, in contrast with a previous study [17] in which subjects measured symptoms every day until the end of the second menstrual cycle. The scores for the 21 DRSP items range from 1 (no) to 6 (extremely serious) and PMS is diagnosed when the total Criteria A score (21 items) is higher than 50.

Health-related behaviors: The health-related behaviors were measured by smoking, drinking, eating habits and nutrient intake. The questionnaire based on the health evaluation index of the 7th Korean National Health and Nutrition Examination Survey (KNHANES) and modified to be suitable to this study, arriving at 14 questions about body shape, weight change/control, drinking, smoking and eating habits. Nutrient intake was measured by the Food Frequency Questionnaire (FFQ) from the 5th and 7th KNHANES. From the fifth survey, the list of food consisted of 63 items under 11 food groups (cereals, pulse/potatoes, meat/poultry, fishes, vegetables, seaweeds, fruits, milk/dairy products, beverages and others). The intake frequency was classified into nine categories (3/2/1 per day, 4–6/2–3/1 per week, 2–3/1 per month and almost none). In the case of fruits, they were classified into seasonal and non-seasonal and added weighted values based on the questionnaire from the 7th KNHANES. The intake amount per serving was classified as small, medium and high based on the standard recommended intake quantity of each food item. The weighted values for these three categories were 0.5, 1 and 2, respectively, based on the database of Korea’s Computer Aided Nutritional Analysis Program (CAN-Pro) 5.0.

General Characteristics: General characteristics were composed of age, marital status, socioeconomic status, grade, height and body weight. Based on the criteria of the Korean Society for the Study of Obesity, BMI (kg/m^2^) was defined less than 18.5 as underweight; 18.5–22.9 as normal; 23–24.9 as overweight; 25–29.9 as obese class I; and above 30.0 as obese class II.

#### 2.4.2. Phase Two

Blood Test: Blood tests were to measure T, sHBG, FBS and insulin. The free androgen index (FAI) was calculated using the formula [FAI = T (nmol/L) × 100/sHBG (nmol/L) and the HOMA-IR was calculated using the formula [HOMA-IR = insulin (μU/mL) × FBS (mg/dL)/450]. In this study, PCOS was assessed by the diagnosis criteria of the National Institutes of Health (NIH) and the National institute of Children’s Health and Disease (NICHD), including oligomenorrhea (or amenorrhea) and biochemical or clinical hyperandrogenism [18]. Irregular menstruation, less than 10 menstrual cycles per year was defined as oligomenorrhea based on a previous study [14] in which sensitivity was 100%, specificity was 78.5%, positive predictive value was 57.8% and negative predictive value was 100%. Hyperandrogenism was defined as the biochemical expression of T ≥ 0.520 ng/mL or FAI ≥ 5.36 [19,20].

Body Composition and BMI: Body composition and BMI were measured with a body composition analyzer (InBody 230, InBody, Seoul, Korea) to estimate muscle mass, body water content, body fat mass, fat-free mass, body fat percentage and body weight. The height was measured with a stadiometer (DS-102, Dong Sahn Jenix, Seoul, Korea) and BMI was categorized the same criteria as in phase one.

### 2.5. Data Collection and Procedure

The survey for menstrual cycles and PMS, health-related behaviors, frequency of food intake and general characteristics was administered online through SurveyMonkey (from 28 August to 22 December 2017). Phase two of the study took place over a total of nine days. Three nurse-licensed researchers at the K University Health Center took blood for testing and measured BMI and body composition. Blood samples were carried to the C Medical Foundation each day for analysis.

### 2.6. Data Analysis

The collected data was analyzed using SAS (Version 9.2, SAS Institute, Cary, NC, USA) and conducted the nutrient analysis using CAN-Pro 5.0 (for Professionals, The Korean Nutrition Society, Seoul, Korea).

Descriptive statistics was used for analyzing menstrual cycle patterns, PMS symptoms and prevalence, PCOS prevalence, body composition, BMI, blood index, health-related behaviors and general characteristics. The differences in the research variables (menstrual cycle patterns, PMS, health-related behaviors, blood index, body composition, BMI and general characteristics) by menstrual regularity and PCOS were analyzed using the t-test, chi-square test and Fisher’s exact test.

## 3. Results

### 3.1. General Characteristics of the Participants

The mean age of the participants was 21.96 years old, with a range from 18 to 29 years. Nearly all women (98.9%) were unmarried and 73.4% (339 women) classified themselves as middle income. Most of the participants (84.4%) were undergraduate students. In terms of BMI, 71.0% (328 women) were normal; 16.2% (75 women) were underweight; 9.1% (42 women) were overweight; 3.3% (15 women) were obese class I; and 0.4% (2 women) were obese class II, as seen in Table 1.

### 3.2. Menarche Age and Menstrual Cycle Patterns

The mean menarche age was 12.36 years old. 51.1% of participants reported regular and 48.9% reported irregular menstruation. For menarche age by menstrual regularity, the mean menarche age of the irregular menstruation women was older than that of regular menstruation women and it was statistically significant difference (*t* = −3.11, *p* = 0.002).

For those who reported regular menstruation, the average number of menstrual cycles for the past year was 11.7; menstrual cycle frequency was 21–35 days for 93.0% (214 women) of participants; less than 21 days for 0.9% (2 women) and more than 36 days for 6.1% (14 women). The mean menstrual period duration was 5.6 days, with 95.2% (218 women) women reporting 2–7 days and 4.8% (11 women) reporting more than 8 days. For menstrual fluid volume, 55.9% (132 women) of the participants had more than 100 points.

In contrast to the above findings, among the women who reported irregular menstruation, the average number of menstrual cycles for the past year was 8.4 times; the shortest menstrual cycle frequency was 28.57 days and the longest was 73.03 days. The shortest mean menstrual period duration was 4.83 days and the longest was 10.44 days. Exactly half (113 women) of the participants scored more than 100 points for menstrual fluid volume (Table 1).

### 3.3. Health-Related Behaviors according to Menstrual Regularity

In the participants’ self-report for body shapes, there was no statistically significant (*p* < 0.05) difference between the women with regular and irregular menstruation (*t* = 8.63, *p* = 0.074), although the women with irregular menstruation showed higher percentages in the categories of very skinny (5.3%), slightly skinny (26.9%) and very obese (2.9%) thandid the women with regular menstruation. There were also no statistically significant dif- ferences in body weight changes (*t* = 3.07, *p* = 0.080) and dietary habits (*t* = 3.24, *p* = 0.072). A higher percentage of the women with irregular menstruation reported body weight and dietary habit changes than did women with regular menstruation (Table 2).

For nutrient intake by menstrual regularity, the women with irregular menstruation showed significantly higher intake levels than did the women with regular menstruation for protein (*t* = −2.24, *p* = 0.026), vitamin K (*t* = −2.44, *p* = 0.015), vitamin C (*t* = −2.81, *p* = 0.005), niacin (*t* = −2.57, *p* = 0.011), vitamin B6 (*t* = −2.79, *p*= 0.006), folic acid (*t*= −2.06, *p* = 0.041), phosphorus (*t* = −2.41, *p* = 0.016), potassium (*t* = −2.85, *p* = 0.005), magnesium (*t*= −3.46, *p*= 0.001), iron (*t* = −2.50, *p* = 0.013), zinc (*t* = −2.95, *p* = 0.003), copper ( *t*= −2.99, *p* = 0.003) and selenium (*t* = −2.14, *p* = 0.033), as seen in Table 3.

### 3.4. PMS Prevalence and Symptoms, Including by Menstrual Regularity

222 women (51.3%) who had total retrospective DRSP scores of more than 50 points were classified as having PMS (Table 4). PMS prevalence was estimated to be 25.5% with a positive predictive value of 52.9%, based on a previous study [11].

According to DSM-IV for PMS, the mean scores of Criteria A and Criteria B were 54.16 and 8.25 points, respectively. Specifically, lethargy had the highest (3.62), followed by anxiety (2.92), appetite changes (2.78), anger (2.74), concentration (2.72), social withdrawal (2.66) and mood swings (2.64) in Criteria A. In Criteria B, productivity had the highest (3.16) (Table 4).

By menstrual regularity, the women with regular menstruation had higher PMS scores than those of their counterparts with irregular menstruation except for control, physical symptoms and interpersonal relationships. However, the differences were not statistically significant (Table 4).

### 3.5. Prevalence of PCOS and Differences in Body Composition and Blood Markers between PCOS and Non-PCOS

In the second phase, 24 women of 88 participants had T ≥ 0.520 ng/mL or FAI ≥ 5.36. Generalizing this result to the entire population (462), the prevalence of PCOS was estimated to be 5.2% based on the PCOS criteria for this study.

For body composition and blood indexes by PCOS, the women with PCOS showed significantly higher for fat-free mass (*t* = 2.04, *p* = 0.049), muscle mass (*t* = 2.13, *p* = 0.041) and body fluid (*t* = 2.07, *p* = 0.047). In addition, they had significantly lower sHBG (*t* = −2.66, *p* = 0.009), higher FAI (*t* = 6.03, *p* < 0.001). However, there were no statistically significant differences in FBS, insulin and HOMA-IR between the women with and without PCOS (Table 5).

## 4. Discussion

In this study, 48.9% of the women reported irregular menstruation. For the women with regular menstruation, the mean number of menstrual cycles in the past year was 11.7, 93.0% showed a normal menstrual cycle frequency (21–35 days) and 95.2% showed a normal menstrual period (2–7 days). Compared the results of the 5th KNHANES (2010–2012), 18.4% of 836 women, aged 19–29, reported irregular menstruation [21]. In Poland, 37.9% of 623 women age 15–19 years, younger than the mean age in this study, reported irregular menstruation and 40.1% of the women with irregular menstruation reported oligomenorrhea [22]. In a subsequent study, 34.5% of 348 women aged 15–25 years reported irregular menstruation and among them 89.4% reported oligomenorrhea, 9.6% reported amenorrhea and 1.0% reported polymenorrhea [23]. In Danish women, 27.0% and 28.5% of the women aged less than 25 years and 25–29 years, respectively, reported irregular menstruation [24]. In this study, it was estimated that 55.9% of the participating women were suspected of heavy menstrual bleeding (HMB) reflected as a PBAC score higher than 100 points. In a study on the self-reported menstrual volume of 19,254 women aged 15–49 years in Japan, 19.4% reported HMB [25]. Another study on the prevalence of HMB in Europe reported 27.2% of all subjects (4506 women aged 18–57 years) having experienced HMB more than two times [26].

In summary, compared with previous findings from both foreign and domestic Korean studies, there were higher percentages of women with both irregular menstruation and HMB in this study. These results are interpreted as follows. First, it suggests that the women who participated in this study had relatively poorer menstrual health. Second, because the subjects themselves were the reporters of their irregular menstruation, there might have been personal differences in how they perceived the scope of criteria. For this reason, a more objective measurement should be considered for further studies. In terms of menstrual bleeding volume, authors of most studies measured it by asking whether the volume was little or a lot. In contrast, it was measured using the PBAC developed to enable relatively objective measurement and widely recommended for clinical trials in this study. However, more studies are needed that incorporate the PBAC in order to further validate it and better reflect the reality. In addition, this study was conducted with the voluntary participation at one university and it could be assumed that a majority of the women who agreed to participate in this study had the intention to check their menstruation health. In other words, women with potential menstrual problems might have been more motivated to participate in this study.

In terms of health-related factors related to menstrual regularity, women with irregular menstruation showed higher percentages of underweight or obese, changes in body weight and dietary habit, albeit not by statistically significant differences. In one study targeting 463 women aged 15–18 years from KNHANES (2009–2013), the results of the nutrient index analysis were not but the intake frequency of coffee and fried foods was significantly high in irregular menstrual women. The researchers in that study also found that earlier menarche age, lack of exercise, high family incomes and high stress levels were associated with irregular menstruation, whereas adequate sleep and less eating-out were more likely to report regular menstruation [27]. Another study on 11,503 Swedish women found that the women who reported binge eating had higher percentages of oligomenorrhea or amenorrhea [28]. Although it is difficult to compare the results of these previous studies directly with those of this study, it implies that menstruation heath and health-related behavior changes including body weight, eating habits or the other lifestyle are related. Therefore, it is suggested that it is needed to investigate the relationship between lifestyle changes and menstruation heath.

This study analyzed the participants’ trends in nutrient intake over the past year was analyzed using the FFQ. The results showed that the women with irregular menstruation had significantly higher intake of protein, niacin, vitamin B_6_, C and K, folic acid, phosphorus, potassium and magnesium, iron, zinc, copper and selenium than did women with regular menstruation. However, it is necessary to take care in interpreting this nutrient intake analysis results by menstrual irregularity. Although it might have captured the participants’ actual nutrient intake status, their overall intakes in all categories were extremely low. Therefore, it is needed to review of FFQ and improve the kinds of food, weighted values for intake amount used in this study.

In this study, 51.3% of the women was classified as having PMS (total retrospective DRSP scores ≥ 50). In contrast, applying the positive predictive value of 52.9% that was measured prospectively and reported in a previous study [11], the estimated PMS prevalence was 25.5%. In the previous study, women whose retrospective DRSP total scores were 50 points or higher took two prospective DRSP measurements and the estimated PMS prevalence was 22.5%. Separately, among the 11 DSM-IV diagnostic symptom areas, lethargy showed the highest score and productivity showed the highest symptom severity scores among the functional impairment areas, similar to this study. The DRSP is recommended as a tool for clinically significant PMS screening. However, there have been few studies using the DRSP and only a few researchers calculated positive predictive value in Korea. Thus more related studies are needed and the positive predictive value that was reported in previous studies for the retrospective measurement of DRSP was applied.

The estimated prevalence of PCOS in this study was 5.2%. In body composition and blood indexes, the PCOS group had significantly higher levels of fat-free mass, muscle mass, body fluid and FAI but lower sHBG levels than did the women without PCOS. Free fat mass and upper-half type body fat distribution (especially, triceps and subscapular region) were reported to increase in women with PCOS [29,30].

For this study, the PCOS prevalence was estimated by the NIH and NICHD diagnostic criteria by generalizing the results for the 24 women, who were found to have total testosterone ≥0.520 ng/mL or FAI ≥ 5.36, out of 88 phase-two subjects, who reported irregular menstruation less than 10 menstrual cycles per year in the first phase, to the entire population of 462 subjects.

This study does, however, have limitations. Because of environmental constraints, this study could not screen out participants who had preceding diseases such congenital adrenal hyperplasia and Cushing syndrome which can cause hyperandrogenism. However, one previous study reported that half of the women with irregular menstruation showed neuroendocrine immaturity as a non-luteal, short/insufficient luteal phase, whereas the other half was implicated in high androgen related to PCOS [31]. In this regard, the PCOS prevalence of this study holds significance as basic data on the actual status of PCOS among Korean women. In one review of studies on worldwide PCOS prevalence [32], the rate estimated with the Rotterdam criteria was 6–21% but that estimated by the NIH diagnostic criteria was 5–10% and the latter was more similar to what was found.

## 5. Conclusions

In conclusion, this study investigated the menstrual cycle patterns of young adult women using specific parameters (volume, frequency, duration and regularity), estimated the prevalence of clinically significant PMS and PCOS and investigated the relationships between the two and health-related behaviors and nutrient intake. The strength of the study was to provide updated basic data about the menstrual health and the clinically significant prevalence of PMS and PCOS of Korean young adult women. However, there are limitations to this study, particularly that the number of participants was small and there were limits in applying diagnostic criteria. Given this, more extensive and sophisticated research needs to be carried out in future.

## Figures and Tables

**Table 1 healthcare-09-00056-t001:** General characteristics, menarche age & menstrual cycle patterns according to menstrual cycle regularity (*n* = 462).

Variables	Total (*n* = 462)		Menstrual Cycle Regularity				*t* or χ2 (p)
			Regular Menstrual Cycle (*n* = 236)		Irregular Menstrual Cycle (*n* = 226)		
	*n* (%)	M ± SD (Min–Max)	*n* (%)	M ± SD (Min—Max)	*n* (%)	M ± SD (Min–Max)	
Age (years)		21.96 ± 2.51 (18–29)		22.05 ± 2.47		21.86 ± 2.54	0.79 (0.431)
Marital status							
Unmarried	457 (98.9)		234 (99.2)		223 (98.7)		0.25 (0.680) ^a^
Married	5 (1.1)		2 (0.8)		3 (1.3)		
Socioeconomic status							
High	61 (13.2)		30 (12.7)		31 (13.7)		0.79 (0.455)
Middle	339 (73.4)		172 (72.9)		167 (73.9)		
Low	62 (13.4)		34 (14.4)		28 (12.4)		
Job							
Under-graduate	390 (84.4)						
Graduate	72 (15.6)						
Height (cm)		161.48 ± 5.12		161.66 ± 4.92		161.29 ± 5.32	0.79 (0.431)
Weight (kg)		53.75 ± 7.28		54.24 ± 7.39		53.24 ± 7.15	1.48 (0.139)
Body mass index		20.59 ± 2.44		20.73 ± 2.43		20.45 ± 2.44	1.21 (0.226)
<18.5	75 (16.2)		38 (16.1)		37 (16.4)		0.79 (0.935) ^a^
18.5–22.9	328 (71.0)		165 (69.9)		163 (72.1)		
23.0–24.9	42 (9.1)		23 (9.8)		19 (8.4)		
25.0–29.9	15 (3.3)		9 (3.8)		6 (2.7)		
≥30.0	2 (0.4)		1 (0.4)		1 (0.4)		
Menarche age (yrs)		12.36 ± 1.52 (8–17)		12.15 ± 1.44 (8–17)		12.58 ± 1.57 (10–17)	−3.11 (0.002)
≤12			90 (38.1)		58 (25.7)		12.75 (0.002) ^a^
13–15			142 (60.2)		154 (68.1)		
≥16			4 (1.7)		14 (6.2)		
No. of menses per year				11.66 ± 1.26 (7–16)		8.40 ± 3.10 (1–20)	
Frequency of menses							
Shortest						28.57 ± 18.23	
Longest						73.03 ± 49.92	
<21 days			2 (0.9)				
21–35 days			214 (93.0)				
>35 days			14 (6.1)				
Missing			6				
Volume of menses		120.40 ± 92.37 (0–642)		124.67 ± 88.15		115.93 ± 96.57	1.02 (0.167)
≥100	246 (53.2)		132 (55.9)		113 (50.0)		1.63 (0.202)
<100	216 (46.8)		104 (44.1)		113 (50.0)		
Duration of menses							
Shortest						4.83 ± 4.05	
Longest						10.44 ± 23.02	
<2 days			0				
2–7 days			218 (95.2)				
≥8 days			11 (4.8)				
Missing			7				

^a^ Fisher exact test; Min–Max, minimum–maximum; M ± SD, mean ± standard deviation.

**Table 2 healthcare-09-00056-t002:** Health-related behaviors according to menstrual cycle regularity (*n* = 462).

Variables	Regular MC (*n* = 236)	Irregular MC (*n* = 226)	χ^2^ (*p*)
	*n* (%)	*n* (%)	
Body image			
Very skinny	7 (3.1)	11 (5.3)	8.63 (0.074) ^a^
Slightly skinny	47 (20.9)	56 (26.9)	
Normal	112 (49.8)	90 (43.3)	
Slightly fat	58 (25.8)	45 (21.6)	
Very fat	1 (0.4)	6 (2.9)	
Missing	11	18	
Body weight change			
No	143 (63.6)	115 (55.3)	3.07 (0.080)
Loss + Gain	82 (36.4)	93 (44.7)	
Missing	11	18	
Body weight loss			
3–6 kg	23 (62.2)	22 (55.0)	2.58 (0.351) ^a^
6–10 kg	13 (35.1)	13 (32.5)	
more than 10 kg	1 (2.7)	5 (12.5)	
Body weight gain			
3–6 kg	30 (66.7)	46 (86.8)	5.84 (0.057) ^a^
6–10 kg	12 (26.7)	5 (9.4)	
more than 10 kg	3 (6.6)	2 (3.8)	
Dietary change			
Yes	65 (28.9)	77 (37.0)	3.24 (0.072)
No	160 (71.1)	131 (63.0)	
Missing	11	18	
Drinking			
less than 4 times per month	195 (86.7)	172 (82.7)	1.32 (0.250)
more than 2 times per week	30 (13.3)	36 (17.3)	
Missing	11	18	
Binge drinking			
less than one time per week	206 (97.6)	190 (96.9)	0.19 (0.765) ^a^
Almost daily	5 (2.4)	6 (3.1)	
Missing	25	30	
Smoking			
Never	204 (90.7)	192 (92.4)	0.89 (0.631) ^a^
not smoke now	13 (5.8)	8 (3.8)	
smoke now	8 (3.5)	8 (3.8)	
Missing	11	18	

^a^ Fisher exact test; MC, menstrual cycle.

**Table 3 healthcare-09-00056-t003:** Nutrients according to menstrual cycle regularity (*n* = 403).

Variables	Regular Menstrual Cycle (*n* = 209)	Irregular Menstrual Cycle (*n* = 194)	*t* (*p*)
	M ± SD	M ± SD	
Calorie (kcal)	1035.18 ± 450.26	1114.60 ± 522.44	−1.63 (0.104)
Carbohydrate (gm)	170.26 ± 79.73	181.45 ± 90.31	−0.32 (0.189)
Fat (gm)	23.42 ± 11.02	25.64 ± 13.76	−1.78 (0.077)
Protein (gm)	36.57 ± 16.88	40.66 ± 19.53	−2.24 (0.026)
Fiber (gm)	12.20 ± 6.75	13.56 ± 7.45	−1.93 (0.055)
Water (gm, mL)	666.51 ± 348.85	719.33 ± 374.65	−1.47 (0.144)
Vitamin A (μg RAE)	220.72 ± 218.65	251.56 ± 287.31	−1.21 (0.229)
Vitamin D (μg)	1.40 ± 1.15	1.46 ± 1.24	−0.50 (0.615)
Vitamin E (mg)	2.77 ± 1.63	3.00 ± 2.02	−1.23 (0.218)
Vitamin K (μg)	22.56 ± 24.25	29.54 ± 32.31	−2.44 (0.015)
Vitamin C (mg)	56.24 ± 38.04	71.05 ± 63.46	−2.81 (0.005)
Thiamin (mg)	0.89 ± 0.40	0.97 ± 0.48	−1.85 (0.065)
Riboflavin (mg)	0.73 ± 0.38	0.80 ± 0.44	−1.70 (0.089)
Niacin (mg)	6.79 ± 3.03	7.67 ± 3.75	−2.57 (0.011)
Vitamin B6 (mg)	0.27 ± 0.16	0.32 ± 0.20	−2.79 (0.006)
Folic acid (μg)	92.96 ± 57.41	106.47 ± 72.90	−2.06 (0.041)
Vitamin B12 (μg)	1.57 ± 1.31	1.79 ± 2.07	−1.25 (0.213)
Pantothenic acid (mg)	0.73 ± 0.56	0.82 ± 0.69	−1.35 (0.178)
Biotin (μg)	1.71 ± 2.10	1.76 ± 2.17	0.23 (0.820)
Calcium (mg)	252.77 ± 155.16	283.53 ± 189.95	−1.77 (0.077)
Phosphorus (mg)	550.64 ± 255.95	619.06 ± 309.17	−2.41 (0.016)
Sodium (mg)	1066.19 ± 680.84	1063.56 ± 643.32	−0.44 (0.968)
Chloride (mg)	95.40 ± 106.78	92.59 ± 106.77	0.26 (0.792)
Potassium (mg)	1338.64 ± 660.74	1550.05 ± 815.73	−2.85 (0.005)
Magnesium (mg)	41.61 ± 26.18	53.12 ± 38.77	−3.46 (0.001)
Iron (mg)	7.46 ± 3.46	8.46 ± 4.44	−2.50 (0.013)
Zinc (mg)	2.88 ± 1.45	3.36 ± 1.79	−2.95 (0.003)
Copper (mg)	0.31 ± 0.17	0.37 ± 0.23	−2.99 (0.003)
Iodine (μg)	42.58 ± 32.06	44.45 ± 31.74	−0.59 (0.557)
Selenium (μg)	28.64 ± 16.37	32.43 ± 18.96	−2.14 (0.033)
Cholesterol (mg)	156.30 ± 104.88	173.99 ± 127.01	−1.52 (0.130)

M ± SD, mean ± standard deviation.

**Table 4 healthcare-09-00056-t004:** Premenstrual symptoms according to menstrual cycle regularity (*n* = 462).

Variables (No. of Items)		Total (*n* = 462)		Menstrual Cycle Regularity				*t* or χ^2^ (*p*)
				Regular Menstrual Cycle (*n* = 236)		Irregular Menstrual Cycle (*n* = 226)		
		*n* (%)	M ± SD	*n* (%)	M ± SD	*n* (%)	M ± SD	
Criteria A								
Depression (3)			2.47 ± 1.22		2.52 ± 1.18		2.42 ± 1.27	0.87 (0.382)
Anxiety (1)			2.92 ± 1.47		2.99 ± 1.43		2.85 ± 1.52	1.02 (0.307)
Irritability (2)			2.64 ± 1.40		2.68 ± 1.39		2.61 ± 1.40	0.48 (0.629)
Anger (2)			2.74 ± 1.21		2.84 ± 1.18		2.64 ± 1.24	1.68 (0.093)
Social withdrawal (1)			2.66 ± 1.43		2.65 ± 1.40		2.66 ± 1.46	−0.07 (0.941)
Concentration (1)			2.72 ± 1.45		2.83 ± 1.39		2.61 ± 1.50	1.62 (0.105)
Lethargy (1)			3.62 ± 1.44		3.67 ± 1.37		3.57 ± 1.52	0.71 (0.476)
Appetite (2)			2.78 ± 1.33		2.87 ± 1.37		2.69 ± 1.28	1.42 (0.157)
Sleep (2)			2.48 ± 1.16		2.50 ± 1.16		2.46 ± 1.16	0.39 (0.699)
Control (2)			1.74 ± 1.10		1.71 ± 1.04		1.76 ± 1.16	−0.50 (0.614)
Physical symptoms (4)			2.51 ± 1.01		2.49 ± 1.01		2.53 ± 1.02	−0.34 (0.735)
	Total of Criteria A		54.16 ± 19.38		54.88 ± 18.86		53.38 ± 19.94	0.80 (0.421)
	≥50	222 (51.3)		118 (52.4)		104 (50.0)		0.26 (0.611)
	<50	211 (48.7)		107 (47.6)		104 (50.0)		
	Missing	29		11		18		
Criteria B								
Productivity (1)			3.16 ± 1.47		3.22 ± 1.49		3.10 ± 1.45	0.86 (0.390)
Social activities (1)			2.91 ± 1.56		2.94 ± 1.58		2.88 ± 1.54	0.38 (0.701)
Interpersonal relationships (1)			2.18 ± 1.33		2.17 ± 1.31		2.18 ± 1.34	−0.07 (0.942)
	Total of Criteria B		8.25 ± 3.88		8.33 ± 3.89		8.16 ± 3.87	0.45 (0.650)

M ± SD, mean ± standard deviation.

**Table 5 healthcare-09-00056-t005:** Body composition and blood indexes among women with PCOS or not (*n* = 88).

Variables	PCOS (*n* = 24)	non-PCOS (*n* = 64)	*t* or χ^2^ (*p*)
	M ± SD	M ± SD	
Body composition			
Fat-free mass (kg)	38.57 ± 4.30	36.62 ± 3.10	2.04 (0.049)
Body fat mass (kg)	18.17 ± 6.42	17.87 ± 12.18	0.15 (0.882)
Muscle (kg)	20.81 ± 2.58	19.59 ± 1.85	2.13 (0.041)
Body fluid (kg)	28.23 ± 3.11	26.79 ± 2.26	2.07 (0.047)
Body mass index (kg/m^2^)	21.91 ± 4.26	20.35 ± 2.10	1.72 (0.096)
Body fat ratio (%)	31.23 ± 6.06	30.06 ± 6.10	0.80 (0.426)
Waist hip ratio	0.83 ± 0.05	0.81± 0.04	1.40 (0.173)
Blood index			
Total testosterone (ng/mL)	0.63 ± 0.15	0.32 ± 0.11	10.40 (<0.001)
Free androgen index	4.94 ± 2.37	1.90 ± 1.12	6.03 (<0.001)
sHBG (nmol/L)	52.57 ± 22.07	68.42 ± 25.89	−2.66 (0.009)
Fasting blood sugar (mg/dL)	90.21 ± 6.29	87.36 ± 8.41	1.51 (0.136)
Insulin (μU/mL)	7.46 ± 3.47	6.73 ± 4.71	0.69 (0.494)
HOMA-IR	1.67 ± 0.84	1.47 ± 1.03	0.86 (0.392)

HOMA-IR, homeostasis model assessment—insulin resistance; PCOS, polycystic ovary syndrome; sHBG, sex hormone binding globulin.

## Data Availability

The data presented in this study are available on request from the corresponding author. The data are not publicly available due to privacy.

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
