# Peer review of "Menstrual Cycle Patterns and the Prevalence of Premenstrual Syndrome and Polycystic Ovary Syndrome in Korean Young Adult Women"

_healthcare, 2021, doi:10.3390/healthcare9010056_

Round 1

Reviewer 1 Report

Dear authors,

I attach the document with suggestions to improve the article.
Best regards.

Author Response

I attached the response to reviewer's comments.

Reviewer 2 Report

The study aimed to investigate the prevalence of Premenstrual Syndrome in a cohort of 88 volunteers subjects, which were assessed for several parameters to describe the statistical correlation to the pathological condition.

Although the study is well structured, the statistical analysis well conducted and presented, the results completely lack of novelty making the manuscript not interesting for the journal readers.

Indeed, several similar studies have been already published in the literature, also considering a larger population.

Author Response

We agree with your opinion. Nonetheless, the basic data for menstrual health, clinically significant PMS prevalence and PCOS prevalence in Korean young adult women was needed to be updated. Especially, the studies about PCOS prevalence of Korean young adult women are paucity. 

Reviewer 3 Report

I recommended the authors to improve presentation of Tables 1 and 2, to better understand their explanation 

Author Response

(The authors gave the same response as above.)

Reviewer 4 Report

Authors should mention what diagnostic criteria they use to confirm PCOS including the latest one (from 2018).

The article is interesting and should be published after minor revision. The most acceptable PCOS criteria are Rotterdam one, 2003 or the latest one published by Teede, 2018. Why authors chose different one? Upper-half type part body fat distribution is very important in PCOS prognosis, ex. triceps fat, it should be discussed due to authors findings. Physical activity without diet changes has no impact according to the latesT literature ex. Pup et al, 2020. It should be also underlined.

Author Response

(The authors gave the same response as above.)

Round 2

Reviewer 2 Report

The paper offers a worthy insight into the updated basic data concerning the menstrual health and the clinically significant prevalence of PMS and PCOS in young adult Korean women; an effort has been made on the part of the authors to fix, in some measure, the article's flaws. More comprehensive criteria have been included in the intro (blood indexes) and within each table, and the inclusion of "health-related behaviors" appears relevant in that respect. New references have also been added, which make for greater cohesion overall.

Some flaws can still be found in the language: "...more similar to what it was found" (line 347), "...and it was applied a positive predictive value that was reported in previous studies" (line 328), in addition to minor typos (e.g. line 257: replace the capitalized T following comma: the prevalence).

In its current form, the article should be deemed fit for publication.

Author Response

1. Some flaws can still be found in the language: "...more similar to what it was found" (line 347), "...and it was applied a positive predictive value that was reported in previous studies" (line 328)

Response 1: We checked the commented sentences and revised according to grammar.

 Line No. 299, 307, 345, 326-327

2. Some flaws can still be found in the language: minor typos (e.g. line 257: replace the capitalized T following comma: the prevalence).

Response 2: We checked and revised the commented sentences.

Line No. 255